

# Fast and exact fixed-radius neighbor search based on sorting

Xinye Chen[1] and Stefan Güttel[2]

[1] Charles University Prague, Prague, Czech Republic
[2] University of Manchester, Manchester, United Kingdom

## ABSTRACT

Fixed-radius near neighbor search is a fundamental data operation that retrieves all data points within a user-specified distance to a query point. There are efficient algorithms that can provide fast approximate query responses, but they often have a very compute-intensive indexing phase and require careful parameter tuning. Therefore, exact brute force and tree-based search methods are still widely used. Here we propose a new fixed-radius near neighbor search method, called SNN, that significantly improves over brute force and tree-based methods in terms of index and query time, provably returns exact results, and requires no parameter tuning. SNN exploits a sorting of the data points by their first principal component to prune the query search space. Further speedup is gained from an efficient implementation using high-level basic linear algebra subprograms (BLAS). We provide theoretical analysis of our method and demonstrate its practical performance when used stand-alone and when applied within the DBSCAN clustering algorithm.

# INTRODUCTION

This work is concerned with the retrieval of nearest neighbors, a fundamental data operation. Given a data point, this operation aims at finding the most similar data points using a predefined distance function. Nearest neighbor search has many applications in computer science and machine learning, including object recognition (*Philbin et al., 2007*; *Nister & Stewenius, 2006*), image descriptor matching (*Silpa-Anan & Hartley, 2008*), time series indexing (*Keogh & Ratanamahatana, 2005*; *Chakrabarti et al., 2002*; *Yagoubi et al., 2020*), clustering (*Ester et al., 1996*; *Campello, Moulavi & Sander, 2013*; *Gallego et al., 2018*; *Alshammari, Stavrakakis & Takatsuka, 2021*; *Gallego, Rico-Juan & Valero-Mas, 2022*; *Li et al., 2020*), particle simulations (*Groß, Köster & Krüger, 2019*), molecular modeling (*Galvelis & Sugita, 2017*), pose estimation (*Shakhnarovich, Viola & Darrell, 2003*), computational linguistics (*Kaminska, Cornelis & Hoste, 2021*), and information retrieval (*Geng et al., 2008*; *Wang et al., 2015*).

There are two main types of nearest neighbor (NN) search: k-nearest neighbor and fixed-radius near neighbor search. Fixed-radius NN search, also referred to as radius query, aims at identifying all data points within a given distance from a query point; see *Bentley (1975b)* for a historical review. The most straightforward way of finding nearest neighbors is *via* a linear search through the whole database, also known as exhaustive or

Corresponding author
Stefan Güttel,
stefan.guettel@manchester.ac.uk

brute force search. Though considered inelegant, it is still widely used, *e.g.*, in combination with GPU acceleration (*Garcia, Debreuve & Barlaud, 2008*).

Existing NN search approaches can be broadly divided into exact and approximate methods. In many applications, approximate methods are an effective solution for performing fast queries while allowing for a small loss. Well-established approximate NN search techniques include randomized $k$-d trees (*Silpa-Anan & Hartley, 2008*), hierarchical k-means (*Nister & Stewenius, 2006*), locality sensitive hashing (*Indyk & Motwani, 1998*), HNSW (*Malkov & Yashunin, 2020*), and ScaNN (*Guo et al., 2020*). Considerable drawbacks of most approximate NN search algorithms are their potentially long indexing time and the need for the tuning of additional hyperparameters such as the trade-off between recall *vs* index and query time. Furthermore, to the best of our knowledge, all approximate NN methods for which open-source implementations (such as *Guo et al., 2020*; *Bernhardsson, 2023*; *Dong, Moses & Li, 2011*; *Malkov & Yashunin, 2020*) are available only address the k-nearest neighbor problem, not the fixed-radius problem discussed here.

In this article we introduce a new *exact* approach to fixed-radius NN search based on sorting, referred to as SNN for short. Some of the appealing properties of SNN are

1. *simplicity:* SNN has no hyperparameters except for the necessary search radius
2. *exactness:* SNN is guaranteed to return all data points within the search radius
3. *speed:* SNN demonstrably outperforms other exact NN search algorithms like, *e.g.*, methods based on tree structures
4. *flexibility:* the low indexing time of SNN makes it applicable in an online streaming setting.

The rest of this article is organized as follows. In "Related Work", we provide a brief review of existing work on NN search. In "Sorting-Based NN Search", we introduce our sorting-based NN method, detailing its indexing and query phases. "Computational Considerations" contains computational considerations regarding the efficient implementation of SNN and its behavior in floating-point arithmetic. Theoretical performance analysis is provided in "Theoretical Analysis". "Experimental Evaluation" contains performance comparisons of our algorithm to other state-of-the-art NN methods, as well as an application to DBSCAN clustering. We then conclude in "Conclusions". (A preprint of this work is available on arXiv (*Chen & Güttel, 2023*)).

## RELATED WORK

NN search methods can broadly be classified into approximate or exact methods, depending on whether they return exact or approximate answers to queries (*Cayton & Dasgupta, 2007*). It is widely accepted that for high-dimensional data there are no exact NN search methods which are asymptotically more efficient than exhaustive search; see, *e.g.*, (*Muja, 2013*, Chap. 3) and *Francis-Landau & Durme (2019)*. Exact NN methods based on $k$-d tree (*Bentley, 1975a*; *Friedman, Bentley & Finkel, 1977*), balltree (*Omohundro, 1989*), VP-tree (*Yianilos, 1993*), cover tree (*Beygelzimer, Kakade & Langford, 2006*), and RP tree (*Dasgupta & Sinha, 2013*) only perform well on low-dimensional data.

This shortcoming is often referred to as the curse of dimensionality (*Indyk & Motwani, 1998*). However, note that negative asymptotic results do not rule out the possibility of algorithms and implementations that perform significantly (by orders of magnitude) faster than brute force search in practice, even on real-world high-dimensional data sets.

To speedup NN search, modern approaches generally focus on two aspects, namely indexing and sketching. The indexing aims to construct a data structure that prunes the search space for a given query, hopefully resulting in fewer distance computations. Sketching, on the other hand, aims at reducing the cost of each distance computation by using a compressed approximate representation of the data.

The most widely used indexing strategy is space partitioning. Some of the earliest approaches are based on tree structures such as $k$-d tree (*Bentley, 1975a*; *Friedman, Bentley & Finkel, 1977*), balltree (*Omohundro, 1989*), VP-tree (*Yianilos, 1993*), and cover tree (*Beygelzimer, Kakade & Langford, 2006*). The tree-based methods are known to become inefficient for high-dimensional data. One of the remedies are randomization (*e.g.*, *Dasgupta & Sinha, 2013*; *Ram & Sinha, 2019*) and ensembling (*e.g.*, the FLANN nearest neighbor search tool by *Muja & Lowe (2009)*, which empirically shows competitive performance against approximate NN methods). Another popular space partitioning method is locality-sensitive hashing (LSH); see, *e.g.*, *Indyk & Motwani (1998)*. LSH leverages a set of hash functions from the locality-sensitive hash family and it guarantees that similar queries are hashed into the same buckets with higher probability than less similar ones. This method was originally introduced for the binary Hamming space by *Indyk & Motwani (1998)*, and it was later extended to the Euclidean space (*Datar et al., 2004*). In *Bawa, Condie & Ganesan (2005)* a self-tuning index for LSH based similarity search was introduced. A partitioning approach based on neural networks and LSH was proposed in *Dong et al. (2020)*. Another interesting method is GriSPy (*Chalela et al., 2021*), which performs fixed-radius NN search using regular grid search—to construct a regular grid for the index—with the possibility of working with periodic boundary conditions. This method, however, has high memory demand because the grid computations grow exponentially with the space dimension.

This article focuses on exact fixed-radius NN search. The implementations available in the most widely used scientific computing environments are all based on tree structures, including `findNeighborsInRadius` in MATLAB (*The MathWorks Inc., 2022*), `NearestNeighbors` in *scikit-learn* (*Pedregosa et al., 2011*), and `spatial` in *SciPy* (*Virtanen et al., 2020*). This is in contrast to our SNN method introduced below which does not utilise any tree structures.

## SORTING-BASED NN SEARCH

Suppose we have $n$ data points $p_1, \ldots, p_n \in \mathbb{R}^d$ (represented as column vectors) and $d \ll n$. The fixed-radius NN problem consists of finding the subset of data points that is closest to a given query point $q \in \mathbb{R}^d$ (may be out-of-sample) with respect to some distance metric. Throughout this article, the vector norm $|| \cdot || = || \cdot ||_2$ is the Euclidean one, though it also possible to identify nearest neighbors with other distances such as

- *cosine distance:* assuming normalized data (with $||u|| = ||v|| = 1$), the cosine distance is

$$\text{cdist}(u, v) = 1 - \cos(\theta) = 1 - \frac{u^T v}{||u|| ||v||} = 1 - u^T v \in [0, 2].$$

Hence, the cosine distance can be computed from the Euclidean distance *via*

$$2\text{cdist}(u, v) = 2 - 2u^T v = u^T u - 2u^T v + v^T v = ||u - v||^2.$$

- *angular distance:* the angular distance $\theta \in [0, \pi]$ between two normalized vectors $u, v$ satisfies

$$\theta \leq \alpha \quad \text{if and only if} \quad ||u - v||^2 \leq 2 - 2\cos(\alpha).$$

  Therefore, closest angle neighbors can be identified *via* Euclidean distance.

- *maximum inner product similarity* (see, *e.g.*, *Bachrach et al., 2014*): for not necessarily normalized vectors we can consider the transformed data points

$$\tilde{p}_i = \left[ \sqrt{\xi^2 - ||p_i||^2}, p_i^T \right]^T \quad \text{with } \xi := \max_i ||p_i|| \text{ and the transformed query point}$$

  $\tilde{q} = [0, q^T]^T$. Then

$$||\tilde{p}_i - \tilde{q}||^2 = ||\tilde{p}_i||^2 + ||\tilde{q}||^2 - 2\tilde{p}_i^T \tilde{q} = \xi^2 + ||q||^2 - 2p_i^T q \geq 0.$$

  Since $\xi$ and $q$ are independent of the index $i$, we have $\text{argmin}_i ||\tilde{p}_i - \tilde{q}||^2 = \text{argmax}_i p_i^T q$.

- *Manhattan distance:* since $||p_i - q||_2 \leq ||p_i - q||_1$, any points satisfying $||p_i - q||_1 > R$ must necessarily satisfy $||p_i - q||_2 > R$. Hence, the sorting-based exclusion criterion proposed in section 3.2 to prune the query search space can also be used for the Manhattan distance.

Our algorithm, called SNN, will return the required indices of the nearest neighbors in the Euclidean norm, and can also return the corresponding distances if needed. SNN uses three essential ingredients to obtain its speed. First, a sorting-based exclusion criterion is used to prune the search space for a given query. Second, pre-calculated dot products of the data points allow for a reduction of arithmetic complexity. Third, a reformulation of the distance criterion in terms of matrices (instead of vectors) allows for the use of high-level basic linear algebra subprograms (BLAS, *Blackford et al., 2002*). In the following, we explain these ingredients in more detail.

### Indexing

Before sorting the data, all data points are centered by subtracting the empirical mean value of each dimension:

$$x_i := p_i - \text{mean}(\{p_j\}).$$

This operation will not affect the pairwise Euclidean distance between the data points and can be performed in $O(dn)$ operations, *i.e.*, with linear complexity in $n$. We then compute the first principal component $v_1 \in \mathbb{R}^d$, *i.e.*, the vector along which the data $\{x_i\}$

exhibits largest empirical variance. This vector can be computed by a thin singular value decomposition of the tall-skinny data matrix $X := [x_1, \ldots, x_n]^T \in \mathbb{R}^{n \times d}$,

$$X = U\Sigma V^T, \tag{1}$$

where $U \in \mathbb{R}^{n \times d}$ and $V \in \mathbb{R}^{d \times d}$ have orthonormal columns and $\Sigma = \text{diag}(\sigma_1, \ldots, \sigma_d) \in \mathbb{R}^{d \times d}$ is a diagonal matrix such that $\sigma_1 \geq \sigma_2 \geq \cdots \geq \sigma_d \geq 0$. The principal components are given as the columns of $V = [v_1, \ldots, v_d]$ and we require only the first column $v_1$. The score of a point $x_i$ along $v_1$ is

$$\alpha_i := x_i^T v_1 = (e_i^T X) v_1 = (e_i^T U\Sigma V^T) v_1 = e_i^T u_1 \sigma_1,$$

where $e_i$ denotes the $i$-th canonical unit vector in $\mathbb{R}^n$. In other words, the scores $\alpha_i$ of all points can be read off from the first column of $U = [u_1, \ldots, u_d]$ times $\sigma_1$. The computation of the scores using a thin SVD requires $O(nd^2)$ operations and is therefore linear in $n$.

The next (and most important) step is to order all data points $x_i$ by their $\alpha_i$ scores; that is,

$$(x_i) := \text{sort}(\{x_i\})$$

so that $\alpha_1 \leq \alpha_2 \leq \cdots \leq \alpha_n$ with each $\alpha_i = x_i^T v_1$. This sorting will generally require a time complexity of $O(n \log n)$ independent of the data dimension $d$. We also precompute the squared-and-halved norm of each data point, $\overline{x}_i = (x_i^T x_i)/2$ for $i = 1, 2, \ldots, n$. This is of complexity $O(nd)$, *i.e.*, again linear in $n$.

All these computations are done exactly once in the indexing phase and only the scores $[\alpha_i]$, the numbers $[\overline{x}_i]$, and the single vector $v_1$ need to be stored. See Algorithm 1 for a summary.

## Query

Given a query point $q$ and user-specified search radius $R$, we want to retrieve all data points $p_i$ satisfying $\|p_i - q\| \leq R$. Figure 1 illustrates our approach. We first compute the mean-centered query $x_q := q - \text{mean}(\{p_j\})$ and the corresponding score $\alpha_q := x_q^T v_1$. By utilizing the Cauchy–Schwarz inequality, we have

$$|\alpha_i - \alpha_q| = |v_1^T x_i - v_1^T x_q| \leq \|x_i - x_q\|. \tag{2}$$

Since we have sorted the $x_i$ such that $\alpha_1 \leq \alpha_2 \leq \cdots \leq \alpha_n$, the following statements are true:

if $\alpha_q - \alpha_{j_1} > R$ for some $j_1$, then $\|x_i - x_q\| > R$ for all $i \leq j_1$;

if $\alpha_{j_2} - \alpha_q > R$ for some $j_2$, then $\|x_i - x_q\| > R$ for all $i \geq j_2$.

As a consequence, we only need to consider *candidates* $x_i$ whose indices are in $J := \{j_1 + 1, j_1 + 2, \ldots, j_2 - 1\}$ and we can determine the smallest subset by finding the largest $j_1$ and smallest $j_2$ satisfying the above statements, respectively. As the $\alpha_i$ are sorted, this can be achieved *via* binary search in $O(\log n)$ operations. Note that the indices in $J$ are

---

**Algorithm 1** SNN index.

1: **Input:** Data matrix $P = [p_1, p_2, \ldots, p_n]^T \in \mathbb{R}^{\times}$

2: Compute $\mu := \text{mean}(\{p_j\})$

3: Compute the mean-centered matrix $X$ with rows $x_i := p_i - \mu$

4: Compute the singular value decomposition of $X = U\Sigma V^T$

5: Compute the sorting keys $\alpha_i = x_i^T v_1$ for $i = 1, 2, \ldots, n$

6: Sort data points $X$ such that $\alpha_1 \leq \alpha_2 \leq \cdots \leq \alpha_n$

7: Compute $\overline{x}_i = (x_i^T x_i)/2$ for $i = 1, 2, \ldots, n$

8: **Return:** $\mu, X, v_1, [\alpha_i], [\overline{x}_i]$

---

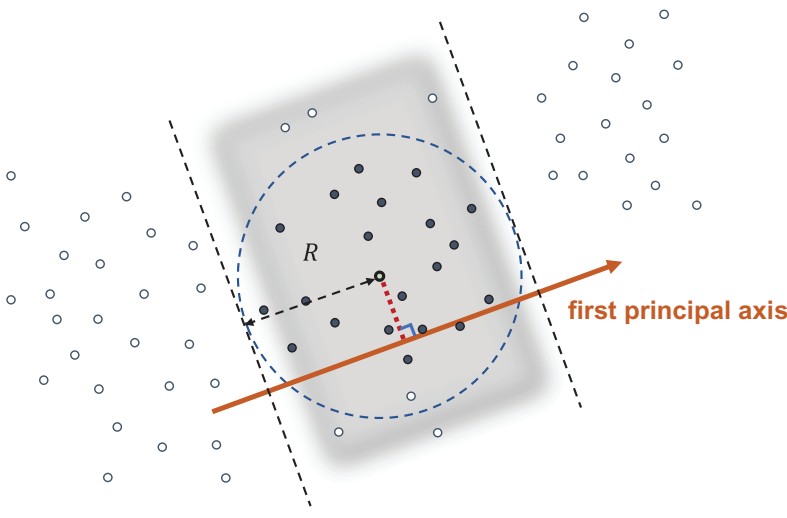

**Figure 1 Query with radius $R$.** The data points in the shaded band have their first principal coordinate within a distance $R$ from the first principal coordinate of the query point, and hence are NN candidates. All data points are sorted so that all candidates have continuous indices.

---

**Algorithm 2** SNN query.

1: **Input:** Query vector $q$; user-specified radius $R$; output from Algorithm 1

2: Compute $x_q := q - \mu$

3: Compute the sorting score of $x_q$, i.e., $\alpha_q := x_q^T v_1$

4: Select candidate index range $J$ so that $|\alpha_j - \alpha_q| \leq R$ for all $j \in J$

5: Compute $d := \overline{x}(J) - X(J, :)^T x_q$ using the precomputed $\overline{x} = [\overline{x}_i]$

6: **Return:** Points $x_j$ with $d_j \leq (R^2 - x_q^T x_q)/2$ according to Eq. (4)

---

continuous integers, and hence it is memory efficient to access $X(J, :)$, the submatrix of $X$ whose row indices are in $J$. This will be important later.

Finally, we filter all data points in the reduced set $X(J, :)$, retaining only those data points whose distance to the query point $x_q$ is less or equal to $R$, i.e., points satisfying $||x_j - x_q||^2 \leq R^2$. The query phase is summarized in Algorithm 2.

## COMPUTATIONAL CONSIDERATIONS

The compute-intensive step of the query procedure is the computation of

$$||x_j - x_q||^2 = (x_j - x_q)^T(x_j - x_q) \tag{3}$$

for all vectors $x_j$ with indices $j \in J$. Assuming that these vectors have $d$ features, one evaluation of Eq. (3) requires $3d - 1$ floating point operations (flop): $d$ flop for the subtractions, $d$ flop for the squaring, and $d - 1$ flop for the summation. In total, $|J|(3d - 1)$ flop are required to compute all $|J|$ squared distances. We can equivalently rewrite (Eq. (3)) as $||x_j - x_q||^2 = x_j^T x_j + x_q^T x_q - 2x_j^T x_q$ and instead verify the radius condition as

$$\frac{1}{2}x_j^T x_j - x_j^T x_q \leq \frac{R^2 - x_q^T x_q}{2}. \tag{4}$$

This form has the advantage that all the squared-and-halved norms $\overline{x_j} = (x_j^T x_j)/2$ ($i = 1, 2, \ldots, n$) have been precomputed during the indexing phase. Hence, in the query phase, the left-hand side of Eq. (4) can be evaluated for all $|J|$ points $x_j$ using only $2d|J|$ flop: $(2d - 1)|J|$ for the inner products and $|J|$ subtractions.

Merely counting flop, Eq. (4) saves about 1/3 of arithmetic operations over (Eq. (3)). An additional advantage results from the fact that all inner products in Eq. (4) can be computed as $X(J, :)^T x_q$ using level-2 BLAS matrix-vector multiplication (gemv), resulting in further speedup on modern computing architectures. If multiple query points are given, say $x_q^{(1)}, \ldots, x_q^{(\ell)}$, a level-3 BLAS matrix-matrix multiplication (gemm) evaluates $X(J, :)^T[x_q^{(1)}, \ldots, x_q^{(\ell)}]$ in one go, where $J$ is the union of candidates for all $\ell$ query points.

One may be concerned that the computation using Eq. (4) incurs more rounding error than the usual Formula (3). We now prove that this is not the case. First, note that division or multiplication by 2 does not incur rounding error. Using the standard model of floating point arithmetic, we have $fl(a \circ b) = (a \circ b)(1 \pm \delta)$ for any elementary operation $\circ \in \{+, -, \times, /\}$, where $0 \leq \delta \leq u$ with the unit roundoff $u$ (*Higham, 2002*, Chap. 1). Suppose we have two vectors $x$ and $y$ where $x_i$ and $y_i$ denote their respective coordinates. Then computing

$$s_d := \sum_{i=1}^{d} (x_i - y_i)^2 = (x - y)^T(x - y)$$

in floating point arithmetic amounts to evaluating

$$\hat{s}_1 = fl((x_1 - y_1)^2) = fl((x_1 - y_1))^2 \cdot (1 \pm \delta) = (x_1 - y_1)^2(1 \pm \delta)^3,$$
$$\hat{s}_2 = fl(\hat{s}_1 + (x_2 - y_2)^2) = (\hat{s}_1 + (x_2 - y_2)^2(1 \pm \delta)^3) \cdot (1 \pm \delta)$$
$$= (x_1 - y_1)^2(1 \pm \delta)^4 + (x_2 - y_2)^2(1 \pm \delta)^4, \text{ and so on.}$$

Continuing this recursion we arrive at

$$\hat{s}_d = (x_1 - y_1)^2(1 \pm \delta)^{d+2} + (x_2 - y_2)^2(1 \pm \delta)^{d+2} + (x_3 - y_3)^2(1 \pm \delta)^{d+1}$$
$$+ \cdots + (x_d - y_d)^2(1 \pm \delta)^4.$$

Assuming $ju < 1$ and using (*Higham, 2002*, Lemma 3.1) we have

$$(1 \pm \delta)^j = 1 + \theta_j, \quad \text{where } |\theta_j| \leq \frac{ju}{1 - ju} := \gamma_j.$$

Hence,

$$\begin{aligned}
&|(x - y)^T(x - y) - fl((x - y)^T(x - y))| \\
&\leq |\theta_{d+2}(x_1 - y_1)^2 + \theta'_{d+2}(x_2 - y_2)^2 + \theta_{d+1}(x_3 - y_3)^2 + \\
&\quad \cdots + \theta_4(x_d - y_d)^2| \\
&\leq |\theta_{d+2}|(x_1 - y_1)^2 + |\theta'_{d+2}|(x_2 - y_2)^2 + |\theta_{d+1}|(x_3 - y_3)^2 + \\
&\quad \cdots + |\theta_4|(x_d - y_d)^2| \\
&\leq \gamma_{d+2}(x - y)^T(x - y),
\end{aligned}$$

showing that the left-hand side of Eq. (3) can be evaluated with high relative accuracy.

A very similar calculation can be done for the formula

$$x^T x + y^T y - 2x^T y = s_d,$$

the expression that is used to derive (Eq. (4)). Using the standard result for inner products (*Higham, 2002*, eq. (3.2))

$$\begin{aligned}
fl(x^T y) = x_1 y_1(1 \pm \delta)^d + x_2 y_2(1 \pm \delta)^d + x_3 y_3(1 \pm \delta)^{d-1} \\
+ \cdots + x_d y_d(1 \pm \delta)^2,
\end{aligned}$$

one readily derives

$$|(x - y)^T(x - y) - fl(x^T x + y^T y - 2x^T y)| \leq \gamma_{d+2}(x - y)^T(x - y),$$

the same bound on the relative accuracy of floating-point evaluation as obtained for (3).

## THEORETICAL ANALYSIS

The efficiency of the SNN query in Algorithm 2 is dependent on the number of pairwise distance computations that are performed in Step 5, depending on the size of the index set $|J|$. If the index set $J$ is the full $\{1, 2, \ldots, n\}$, then the algorithm reduces to exhaustive search over the whole dataset $\{x_1, x_2, \ldots, x_n\}$, which is undesirable. For the algorithm to be most efficient, $|J|$ would exactly coincide with the indices of data points $x_i$ that satisfy $||x_i - x_q|| \leq R$. In practice, the index set $J$ will be somewhere in between these two extremes. Thus, it is natural to ask: *How likely is it that $|\alpha_i - \alpha_q| \leq R$, yet $||x_i - x_q|| > R$?*

First note that, using the singular value decomposition (Eq. (1)) of the data matrix $X$, we can derive an upper bound on $||x_i - x_q||$ that complements the lower bound (Eq. (2)). Using that $x_i^T = e_i^T X = e_i^T U \Sigma V^T$, where $e_i \in \mathbb{R}^n$ denotes the $i$th canonical unit vector, and denoting the elements of $U$ by $u_{ij}$, we have

$$\begin{aligned}
||x_i - x_q||^2 &= |\alpha_i - \alpha_q|^2 + ||[(u_{i2} - u_{q2}), \ldots, (u_{id} - u_{qd})]\hat{\Sigma}||^2 \\
&\leq |\alpha_i - \alpha_q|^2 + ||u_i - u_q||^2 \cdot ||\hat{\Sigma}||^2 \\
&\leq |\alpha_i - \alpha_q|^2 + 2\sigma_2^2
\end{aligned}$$

with $\hat{\Sigma} = \begin{bmatrix} \sigma_2 & & \\ & \ddots & \\ & & \sigma_d \end{bmatrix}$. Therefore,

$$|\alpha_i - \alpha_q|^2 \leq ||x_i - x_q||^2 \leq |\alpha_i - \alpha_q|^2 + 2\sigma_2^2 \qquad (5)$$

and the gap in these inequalities depends on $\sigma_2$, the second singular value of $X$. Indeed, if $\sigma_2 = 0$, then all data points $x_i$ lie on a straight line passing through the origin and their distances correspond exactly to the difference in their first principal coordinates. This is a best-case scenario for Algorithm 2 as all candidates $x_j, j \in J$, found in Step 4 are indeed also nearest neighbors. If, on the other hand, $\sigma_2$ is relatively large compared to $\sigma_1$, the gap in the inequalities (Eq. (5)) becomes large and $|\alpha_i - \alpha_q|$ may be a crude underestimation of the distance $||x_i - x_q||$.

In order to get a qualitative understanding of how the number of distance computations in Algorithm 2 depends on the various parameters (dimension $d$, singular values of the data matrix, query radius $R$, *etc.*), we consider the following model. Let $\{x_i\}_{i=1}^n$ be a large sample of points whose $d$ components are normally distributed with zero mean and standard deviation $[1, s, \ldots, s]$, $s < 1$, respectively. These points describe an elongated "Gaussian blob" in $\mathbb{R}^d$, with the elongation controlled by $s$. In the large data limit $(n \to \infty)$ the singular values of the data matrix $X = [x_1, \ldots, x_n]^T$ approach $\sqrt{n}, s\sqrt{n}, \ldots, s\sqrt{n}$ and the principal components approach the canonical unit vectors $e_1, e_2, \ldots, e_d$. As a consequence, the principal coordinates $\alpha_i = e_1^T x_i$ follow a standard normal distribution, and hence for any $c \in \mathbb{R}$ the probability that $|\alpha_i - c| \leq R$ is given as

$$P_1 = P_1(c, R) = \frac{1}{\sqrt{2\pi}} \int_{c-R}^{c+R} e^{-r^2/2} \mathrm{d}r.$$

On the other hand, the probability that $||x_i - [c, 0, \ldots, 0]^T|| \leq R$ is given by

$$P_2 = P_2(c, R, s, d)$$
$$= \frac{1}{\sqrt{2\pi}} \int_{c-R}^{c+R} e^{-r^2/2} \cdot F\left(\frac{R^2 - (r - c)^2}{s^2}; d - 1\right) \mathrm{d}r, \qquad (6)$$

where $F$ denotes the $\chi^2$ cumulative distribution function. In this model we can think of the point $x_q := [c, 0, \ldots, 0]^T$ as a query point, and our aim is to identify all data points $x_i$ within a radius $R$ of this query point.

Since $||x_i - x_q|| \leq R$ implies that $|\alpha_i - c| \leq R$, we have $P_1 \geq P_2$. Hence, the quotient $P_2/P_1$ can be interpreted as a conditional probability of a point $x_i$ satisfying $||x_i - x_q|| \leq R$ given that $|e_1^T x_i - c| \leq R$, *i.e.*,

$$P = P(||x_i - x_q|| \leq R \mid |e_1^T x_i - c| \leq R) = P_2/P_1.$$

Ideally, we would like this quotient $P = P_2/P_1$ be close to 1, and it is now easy to study the dependence on the various parameters. First note that $P_1$ does not depend on $s$ nor $d$, and hence the only effect these two parameters have on $P$ is *via* the factor

$F\left(\frac{R^2-(r-c)^2}{s^2};d-1\right)$ in the integrand of $P_2$. This term corresponds to the probability that the sum of squares of $d-1$ independent Gaussian random variables with mean zero and standard deviation $s$ is less or equal to $R^2-(r-c)^2$. Hence, $P_2$ and therefore $P$ are monotonically decreasing as $s$ or $d$ are increasing. This is consistent with intuition: as $s$ increases, the elongated point cloud $\{x_i\}$ becomes more spherical and hence it gets more difficult to find a direction in which to enumerate (sort) the points naturally. And this problem gets more pronounced in higher dimensions $d$.

We now show that the "efficiency ratio" $P$ converges to 1 as $R$ increases. In other words, the identification of candidate points $x_j, j \in J$, should become relatively more efficient as the query radius $R$ increases. (Here relative is meant in the sense that candidate points become more likely to be fixed-radius nearest neighbors as $R$ increases. Informally, as $R \to \infty$, all $n$ data points are candidates and also nearest neighbors and so the efficiency ratio must be 1.) First note that for an arbitrarily small $\varepsilon > 0$ there exists a radius $R_1 > 1$ such that $P_1(c, R_1 - 1) > 1 - \varepsilon$. Further, there is a $R_2 > 1$ such that

$$F\left(\frac{R_2^2-(r-c)^2}{s^2};d-1\right) > 1-\varepsilon \quad \text{for all } r \in [c-R_2+1, c+R_2-1].$$

To see this, note that the cumulative distribution function $F$ increases monotonically from 0 to 1 as its first argument increases from 0 to $\infty$. Hence there exists a value $T$ for which $F(t, d-1) > 1-\varepsilon$ for all $t \geq T$. Now we just need to find $R_2$ such that

$$\frac{R_2^2-(r-c)^2}{s^2} \geq T \quad \text{for all } r \in [c-R_2+1, \ c+R_2-1].$$

The left-hand side is a quadratic function with roots at $r = c \pm R_2$, symmetric with respect to the maximum at $r = c$. Hence choosing $R_2$ such that

$$\frac{R_2^2-([c+R_2-1]-c)^2}{s^2} = T, \quad i.e., \quad R_2 = \left(\frac{Ts^2+1}{2}\right)^{1/2},$$

or any value $R_2$ larger than that, will be sufficient. Now, setting $R = \max\{R_1, R_2\}$, we have

$$P_2 \geq \frac{1}{\sqrt{2\pi}} \int_{c-R+1}^{c+R-1} e^{-r^2/2} \cdot F\left(\frac{R^2-(r-c)^2}{s^2};d-1\right) dr \geq (1-\varepsilon)^2.$$

Hence, both $P_1$ and $P_2$ come arbitrarily close to 1 as $R$ increases, and so does their quotient $P = P_2/P_1$.

## EXPERIMENTAL EVALUATION

Our experiments are conducted on a compute server with two Intel Xeon Silver 4114 2.2G processors, 1.5 TB RAM, with operating system Linux Debian 11. All algorithms are forced to run in a single thread with the same settings for fair comparison. We only consider algorithms for which stable Cython or Python implementation are freely available. Our SNN algorithm is implemented in native Python (*i.e.*, no Cython is used), while

**Table 1** The table shows the ratio of returned data points from the synthetic uniformly distributed dataset, relative to the overall number of points $n$, as the query radius $R$ and the dimension $d$ is varied; The ratios confirm that our parameter choices lead to queries over a wide order-in-magnitude variation of query return sizes.

| | $n$ | 2,000 | 4,000 | 6,000 | 8,000 | 10,000 | 12,000 | 14,000 | 16,000 | 18,000 | 20,000 |
|---|---|---|---|---|---|---|---|---|---|---|---|
| Varying $n$ ($d=2$) | $R=0.02$ | 0.1243% | 0.1245% | 0.1232% | 0.1234% | 0.1236% | 0.1236% | 0.1238% | 0.1238% | 0.1232% | 0.1234% |
| | $R=0.05$ | 0.755% | 0.7561% | 0.7506% | 0.7522% | 0.7532% | 0.7508% | 0.7531% | 0.7534% | 0.7511% | 0.751% |
| | $R=0.08$ | 1.852% | 1.87% | 1.879% | 1.871% | 1.882% | 1.879% | 1.879% | 1.871% | 1.881% | 1.875% |
| | $R=0.11$ | 3.434% | 3.46% | 3.433% | 3.449% | 3.44% | 3.459% | 3.454% | 3.44% | 3.452% | 3.453% |
| | $R=0.14$ | 5.487% | 5.42% | 5.443% | 5.456% | 5.438% | 5.454% | 5.442% | 5.432% | 5.449% | 5.417% |
| Varying $n$ ($d=50$) | $R=2.0$ | 0.01732% | 0.01674% | 0.01818% | 0.01763% | 0.01752% | 0.01717% | 0.01734% | 0.01726% | 0.0175% | 0.0174% |
| | $R=2.1$ | 0.07652% | 0.07361% | 0.07286% | 0.07502% | 0.07777% | 0.07414% | 0.07571% | 0.07737% | 0.07486% | 0.07722% |
| | $R=2.2$ | 0.2873% | 0.2843% | 0.2912% | 0.2863% | 0.2879% | 0.2857% | 0.2862% | 0.2903% | 0.2888% | 0.2892% |
| | $R=2.3$ | 0.9608% | 0.9235% | 0.9316% | 0.9184% | 0.9129% | 0.929% | 0.9065% | 0.9195% | 0.9166% | 0.9303% |
| | $R=2.4$ | 2.514% | 2.624% | 2.623% | 2.544% | 2.526% | 2.511% | 2.542% | 2.558% | 2.584% | 2.562% |
| | $d$ | 2 | 32 | 62 | 92 | 122 | 152 | 182 | 212 | 242 | 272 |
| Varying $d$ ($n=10,000$) | $R=0.5$ | 48.11% | 0% | 0% | 0% | 0% | 0% | 0% | 0% | 0% | 0% |
| | $R=2.0$ | 100.0% | 11.08% | 3.9e−05% | 0% | 0% | 0% | 0% | 0% | 0% | 0% |
| | $R=3.5$ | 100.0% | 100.0% | 88.78% | 4.613% | 0.001981% | 0% | 0% | 0% | 0% | 0% |
| | $R=5.0$ | 100.0% | 100.0% | 100.0% | 100.0% | 98.03% | 45.5% | 1.886% | 0.005933% | 2e−06% | 0% |
| | $R=6.5$ | 100.0% | 100.0% | 100.0% | 100.0% | 100.0% | 100.0% | 100.0% | 98.99% | 74.06% | 17.15% |

*scikit-learn*'s (*Pedregosa et al., 2011*) $k$-d tree and balltree NN algorithms, and hence also scikit-learn's DBSCAN method, use Cython for some part of the computation. Numerical values are reported to four significant digits. The code and data to reproduce the experiments in this article can be downloaded from https://github.com/nla-group/snn.

**Near neighbor query on synthetic data**

We first compare $k$-d tree, balltree, and SNN on synthetically generated data to study their dependence on the data size $n$ and the data dimension $d$. We also include two brute force methods, the one in *scikit-learn* (*Pedregosa et al., 2011*) (denoted as brute force 1) and another one implemented by us (denoted as brute force 2) which exploits BLAS level-2 (One might say that brute force 2 is equivalent to SNN without index construction and without search space pruning.). The leaf size for *scikit-learn*'s $k$-d tree and balltree is kept at the default value 40. The $n$ data points are obtained by sampling from the uniform distribution on $[0, 1]^d$.

For the first test we vary the number of data points $n$ (the index size) from 2,000 to 20,000 in increments of 2,000. The number of features is either $d = 2$ or $d = 50$. We then query the nearest neighbors of each data point for varying radius $R$. The ratio of returned data points relative to the overall number of points is listed in Table 1. As expected, this ratio is approximately independent of $n$. We have chosen the radii $R$ so that a good order-of-magnitude variation in the ratio is obtained, in order to simulate queries with small to large returns. The timings of the index and query phases of the various NN algorithms are shown in Fig. 2 (left). Note that the brute force methods do not require the construction of

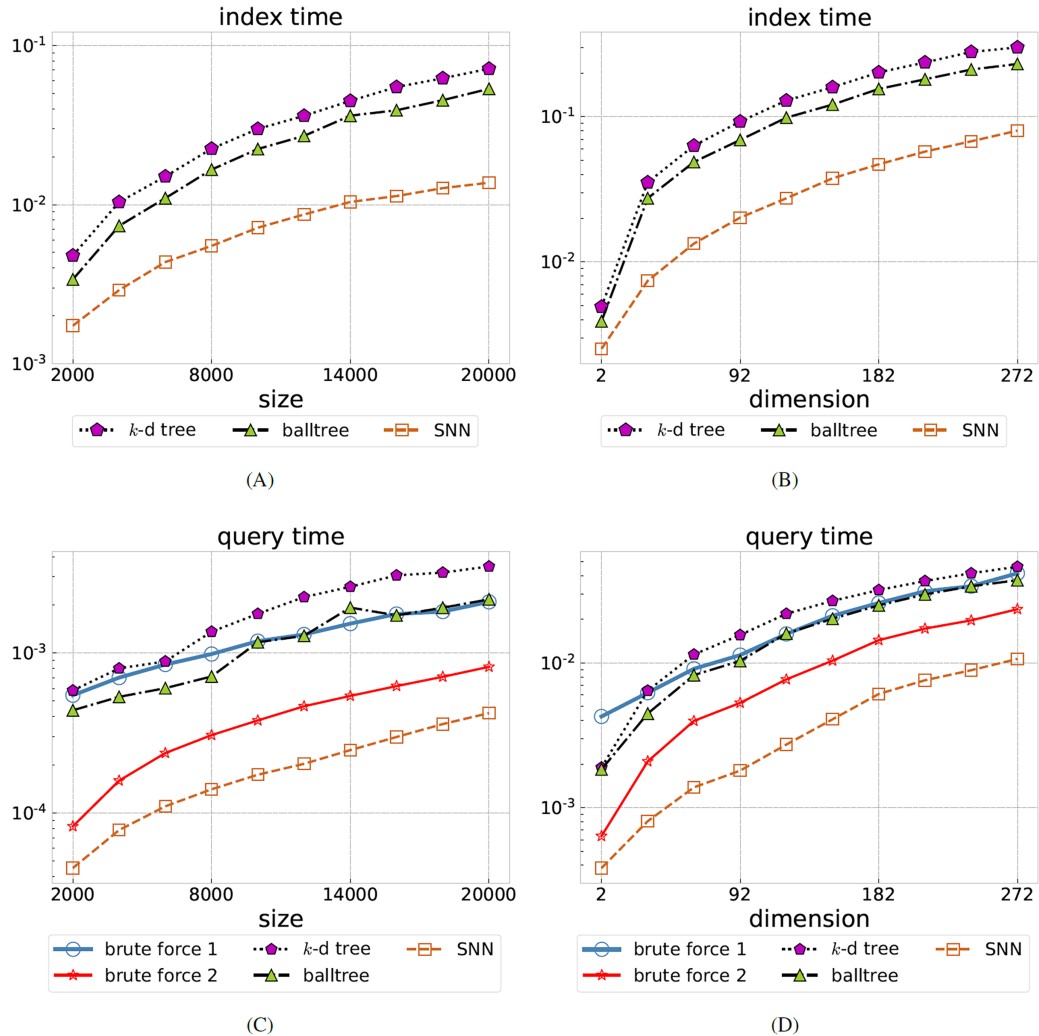

**Figure 2 Comparing SNN to brute force search and tree-based methods.** Total index time (A, B) and average query time (C, D) for the synthetic uniformly distributed dataset, all in seconds, as the data size $n$ is varied (A, C) or the dimension $d$ is varied (B, D). Brute force query methods do not require an index construction, hence are omitted on the left. Our SNN method is the best performer in all cases, in some cases 10 times faster than the best tree-based method (balltree).

an index. Among $k$-d tree, balltree, and SNN, our method has the shortest indexing phase. The query time is obtained as an average over all queries, over the two considered dimensions $d \in \{2, 50\}$, and over all considered radii $R$.SNN performs best, with the average query time being between 5 and 9.7 times faster than balltree (the fastest tree-based method). We have verified that for all methods, when run on the same datasets with the same radius parameter, the set of returned points coincide.

For the second test we fix the number of data points at $n = 10{,}000$ and vary the dimension $d = 2, 32, \ldots, 272$. We perform queries for five selected radii as shown in Table 1. The table confirms that we have a wide variation in the number of returned data

**Table 2 The table shows the ratio of returned data points from the synthetic uniformly distributed dataset, relative to the overall number of points $n$, as the data volumn $n$, query radius $R$ and the dimension $d$ is varied.**

|  |  | $n$ | 1,000 | 2,154 | 4,641 | 10,000 | 21,544 | 46,415 | 100,000 |
|---|---|---|---|---|---|---|---|---|---|
| Varying $n$ ($d = 3$) | $R = 0.05$ | | 0.05% | 0.05% | 0.048% | 0.049% | 0.05% | 0.05% | 0.049% |
| | $R = 0.10$ | | 0.37% | 0.37% | 0.36% | 0.38% | 0.37% | 0.37% | 0.37% |
| | $R = 0.15$ | | 1.2% | 1.2% | 1.2% | 1.2% | 1.2% | 1.2% | 1.2% |
| | $R = 0.20$ | | 2.6% | 2.6% | 2.6% | 2.7% | 2.7% | 2.6% | 2.6% |
| | $R = 0.25$ | | 4.8% | 4.8% | 4.8% | 4.9% | 4.9% | 4.9% | 4.7% |
| | $d$ | | 2 | 3 | 4 | | | | |
| Varying $d$ ($n = 10,000$) | $R = 0.05$ | | 0.75% | 0.05% | 0.0029% | | | | |
| | $R = 0.10$ | | 2.9% | 0.38% | 0.042% | | | | |
| | $R = 0.15$ | | 6.1% | 1.2% | 0.2% | | | | |
| | $R = 0.20$ | | 10% | 2.7% | 0.59% | | | | |
| | $R = 0.25$ | | 15% | 4.9% | 1.3% | | | | |

points relative to the overall number of points $n$, ranging from empty returns to returning all data points. The indexing and query timings are shown in Fig. 2 (right). Again, among $k$-d tree, balltree, and SNN, our method has the shortest indexing phase. The query time is obtained as an average over all $n$ query points and over all considered radii $R$. SNN performs best, with the average query time being between 3.5 and 6 times faster than balltree (the fastest tree-based method).

## Comparison with GriSPy

GriSPy (*Chalela et al., 2021*), perhaps the most recent work on fixed-radius NN search, is an exact search algorithm which claims to be superior over the tree-based algorithms in *SciPy*. GriSPy indexes the data points into regular grids and creates a hash table in which the keys are the cell coordinates, and the values are lists containing the indices of the points within the corresponding cell. As there is an open-source implementation available, we can easily compare GriSPy against SNN. However, GriSPy has a rather high memory demand which forced us to perform a separate experiments with reduced data sizes and dimensions as compared to the ones in the previous "Near Neighbor Query on Synthetic Data".

Again we consider $n$ uniformly distributed data points in $[0, 1]^d$, but now with (i) varying data size from $n = 1,000$ to $100,000$ and averaging the runtime of five different radius queries with $R = 0.05, 0.1, \ldots, 0.25$, and (ii) varying dimension over $d = 2, 3, 4$. The precise parameters and the corresponding ratio of returned data points are listed in Table 2. All queries are repeated 1,000 times and timings are averaged. Both experiments (i) and (ii) use the same query size as the index size.

The index and query timings are illustrated in Fig. 3. We find that SNN indexing is about an order of magnitude faster than GriSPy over all tested parameters. For the experiment (i) where the data size is varied, we find that SNN is up to two orders of

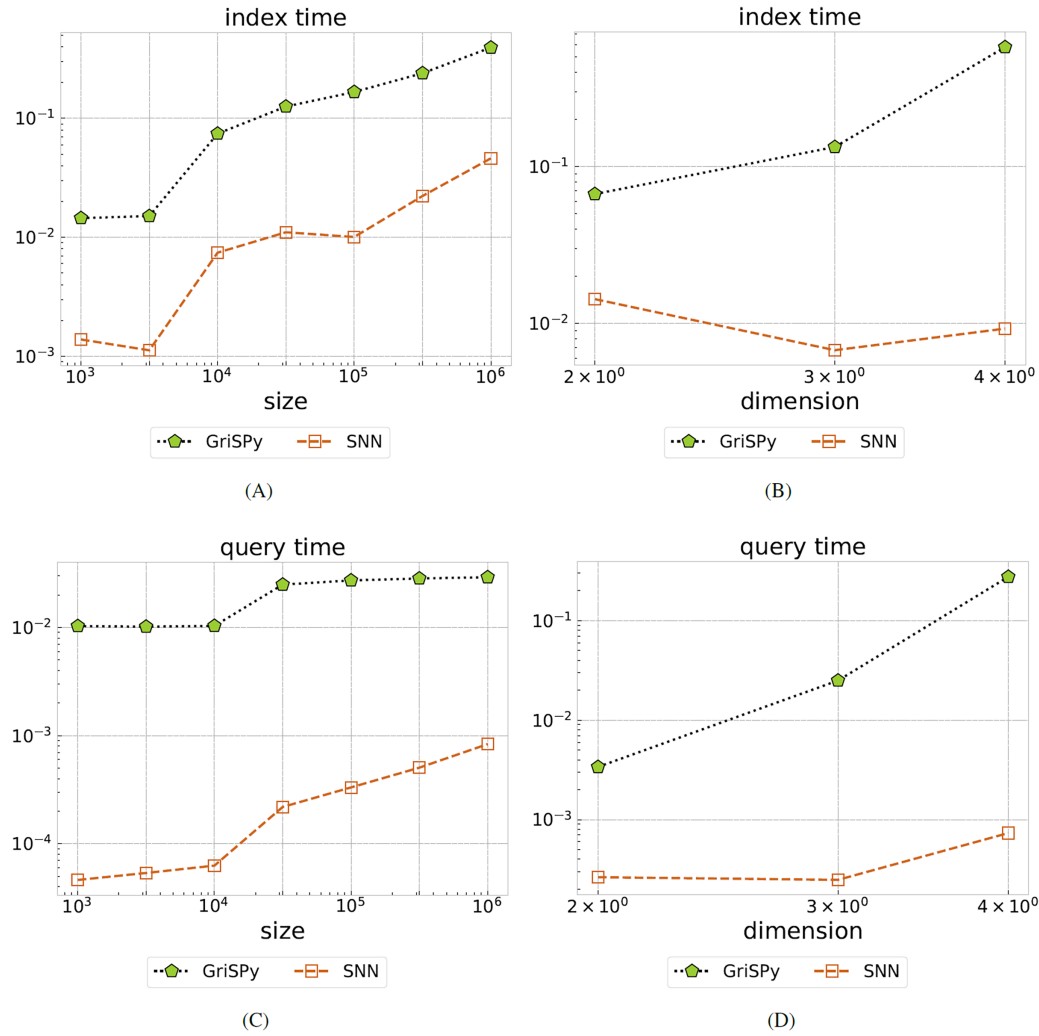

**Figure 3 Comparing GriSPy and SNN.** Total index time (A, B) and average query time (C, D) for on uniformly distributed data, all in seconds, as the data size *n* is varied (A, C) or the dimension *d* is varied (B, D). Our SNN method significantly outperforms GriSPy both in terms of indexing and query runtime.

magnitude faster than GriSPy. For experiment (ii), we see that the SNN query time is more stable than GriSPy with respect to increasing data dimension.

## Near neighbor query on real-world data

We now compare various fixed-radius NN search methods on datasets from the benchmark collection by *Aumüller, Bernhardsson & Faithfull (2020)*: Fashion-MNIST (abbreviated as F-MNIST), SIFT, GIST, GloVe100, and DEEP1B. Each dataset has an index set of *n* points and a separate out-of-sample query set with $n' < n$ points. See Table 3 for a summary of the data.

Table 4 lists the timings for the index construction of the tree-based methods and SNN. For all datasets, SNN is least 5.9 times faster than balltree (the fasted tree-based method).

**Table 3 Summary of the real-world datasets.**

| Dataset | Dimension $d$ | Distance | Index size $n$ | Query size $n'$ | Related reference |
|---------|---------------|----------|----------------|-----------------|-------------------|
| F-MNIST | 784 | Euclidean | 25,000 | 10,000 | *Xiao, Rasul & Vollgraf (2017)* |
| SIFT10K | 128 | Euclidean | 25,000 | 100 | *Lowe (2004)* |
| SIFT1M | 128 | Euclidean | 100,000 | 10,000 | *Lowe (2004)* |
| GIST | 960 | Euclidean | 1,000,000 | 1,000 | *Oliva & Torralba (2004)* |
| GloVe100 | 100 | Angular | 1,183,514 | 10,000 | *Pennington, Socher & Manning (2014)* |
| DEEP1B | 96 | Angular | 9,990,000 | 10,000 | *Yandex & Lempitsky (2016)* |

**Table 4 Index time in milliseconds for fixed-radius NN search on the real-world datasets (rounded to four significant digits).**

| Dataset | $k$-d tree | Balltree | SNN |
|---------|-----------|----------|-----|
| F-MNIST | 9,035 | 7,882 | **1,335** |
| SIFT10K | 720.5 | 662.1 | **79.1** |
| SIFT1M | 3,292 | 2,921 | **179** |
| GIST | 319,400 | 297,900 | **29,140** |
| GloVe100 | 41,210 | 39,800 | **1,549** |
| DEEP1B | 446,000 | 464,100 | **14,730** |

**Note:**
Lower is better and the best values are highlighted in bold.

Significant speedups are gained in particular for large datasets: for the largest dataset DEEP1B, SNN creates its index more than 32 times faster than balltree.

The query times averaged over all $n'$ points from the query set are listed in Table 5. We have included tests over different radii $R$ in order to obtain a good order-of-magnitude variation in the number of returned nearest neighbors relative to the index size $n$, assessing the algorithms over a range of possible scenarios from small to large query returns. See the return ratios $\bar{\upsilon}$ listed in Table 5. Again, in all cases, SNN consistently performs the fastest queries over all datasets and radii. SNN is between about 6 and 14 times faster than balltree (the fastest tree-based method). For the datasets GloVe100 and DEEP1B, SNN displays the lowest speedup of about 1.6 compared to our brute force 2 implementation, indicating that for these datasets the sorting-based exclusion criterion does not significantly prune the search space. (These are datasets for which the angular distance is used, *i.e.*, all data points are projected onto the unit sphere.) For the other datasets, SNN achieves significant speedups between 2.6 and 5.6 compared to brute force 2, owing to effective search space pruning.

## An application to clustering

We now wish to demonstrate the performance gains that can be obtained with SNN using the DBSCAN clustering method (*Campello et al., 2015*; *Jang & Jiang, 2019*) as an example. To this end we replace the nearest neighbor search method in scikit-learn's DBSCAN implementation with SNN. To enure all variants perform the exact same NN queries, we

**Table 5 Query time per data point in milliseconds for real-world data, averaged over $n'$ out-of-sample queries.**

| Dataset | $R$ | $\bar{v}$ | Brute force 1 | Brute force 2 | $k$-d tree | Balltree | SNN |
|---------|-----|-----------|---------------|---------------|------------|----------|------|
| F-MNIST | 800 | 0.01524% | 302.8 | 43.99 | 146.3 | 110.3 | **7.765** |
| | 900 | 0.04008% | 244.4 | 43.96 | 152.2 | 110.7 | **8.602** |
| | 1,000 | 0.09283% | 218.5 | 44.1 | 157.2 | 111.2 | **9.413** |
| | 1,100 | 0.1960% | 217.3 | 44.28 | 160.5 | 111.5 | **10.21** |
| | 1,200 | 0.3818% | 216.2 | 44.32 | 163.3 | 110.8 | **11.18** |
| SIFT10K | 210 | 0.02296% | 19.04 | 4.187 | 15.88 | 12.55 | **1.112** |
| | 230 | 0.04892% | 21.56 | 4.153 | 18.58 | 13.75 | **1.170** |
| | 250 | 0.1147% | 21.24 | 4.546 | 18.35 | 13.15 | **1.458** |
| | 270 | 0.2718% | 22.97 | 4.279 | 19.91 | 14.73 | **1.128** |
| | 290 | 0.5958% | 22.06 | 4.276 | 19.86 | 15.33 | **1.093** |
| SIFT1M | 210 | 0.02661% | 75.82 | 16.10 | 45.71 | 35.11 | **4.525** |
| | 230 | 0.05671% | 78.24 | 16.15 | 46.86 | 38.37 | **4.557** |
| | 250 | 0.1231% | 86.03 | 16.29 | 50.30 | 40.75 | **4.598** |
| | 270 | 0.2663% | 80.13 | 16.17 | 54.13 | 42.76 | **4.660** |
| | 290 | 0.5608% | 69.77 | 16.17 | 58.80 | 44.55 | **4.727** |
| GIST | 0.80 | 0.1430% | 3,955 | 862.2 | 3,144 | 2,160 | **281.5** |
| | 0.85 | 0.1977% | 3,966 | 861.4 | 3,182 | 2,164 | **293.9** |
| | 0.90 | 0.2723% | 3,941 | 861.6 | 3,206 | 2,171 | **305.8** |
| | 0.95 | 0.3762% | 3,817 | 861.7 | 3,223 | 2,178 | **316.8** |
| | 1.00 | 0.5234% | 3,759 | 861.4 | 3,237 | 2,183 | **326.8** |
| GloVe100 | $0.30\pi$ | 0.04506% | 516.9 | 127.3 | 671.5 | 567.5 | **78.38** |
| | $0.31\pi$ | 0.07888% | 514.1 | 126.9 | 673.2 | 561.8 | **79.47** |
| | $0.32\pi$ | 0.1438% | 514.7 | 126.8 | 670.6 | 564.9 | **76.83** |
| | $0.33\pi$ | 0.2755% | 520.1 | 126.5 | 674.9 | 561.0 | **77.27** |
| | $0.34\pi$ | 0.5507% | 522.0 | 127.8 | 674.6 | 562.2 | **77.00** |
| DEEP1B | $0.22\pi$ | 0.04495% | 4,281 | 1,079 | 5,711 | 4,731 | **803.0** |
| | $0.24\pi$ | 0.09332% | 4,229 | 1,065 | 5,677 | 4,704 | **704.8** |
| | $0.26\pi$ | 0.1891% | 4,202 | 1,082 | 5,732 | 4,683 | **719.9** |
| | $0.28\pi$ | 0.3761% | 4,230 | 1,080 | 5,765 | 4,755 | **734.3** |
| | $0.30\pi$ | 0.7341% | 4,274 | 1,084 | 5,644 | 4,810 | **723.1** |

**Note:**
The search radius is $R$ and $\bar{v}$ is the average ratio of returned data points relative to the overall number of data points $n$. Lower is better and the best values are highlighted in bold.

rewrite all batch NN queries into loops of single queries and force all computations to run in a single threat. Except these modifications, DBSCAN remains unchanged and in all case returns exactly the same result when called on the same data and with the same hyperparameters (eps and min_sample).

We select datasets from the UCI Machine Learning Repository (*Dua & Graff, 2017*); see Table 6. All datasets are pre-processed by z-score standardization (*i.e.*, we shift each feature to zero mean and scale it to unit variance). We cluster the data for various choices of DBSCAN's eps parameter and list the measured total runtime in Table 7. The parameter

**Table 6 Clustering datasets in the UCI machine learning repository.**

| Dataset | Size $n$ | Dimension $d$ | #Labels | Related references |
|---|---|---|---|---|
| Banknote | 1,372 | 4 | 2 | *Dua & Graff (2017)* |
| Dermatology | 366 | 34 | 6 | *Dua & Graff (2017)*, *Güvenir, Demiröz & Ilter (1998)* |
| Ecoli | 336 | 7 | 8 | *Dua & Graff (2017)*, *Nakai & Kanehisa (1991, 1992)* |
| Phoneme | 4,509 | 256 | 5 | *Hastie, Tibshirani & Friedman (2009)* |
| Wine | 178 | 13 | 3 | *Forina et al. (1998)* |

**Table 7 Total DBSCAN runtime in milliseconds when different NN search algorithms are used.**

| Dataset | eps | NMI | Brute force | $k$-d tree | Balltree | SNN |
|---|---|---|---|---|---|---|
| Banknote | 0.1 | 0.05326 | 1,914 | 463.9 | 431.9 | **30.20** |
| | 0.2 | 0.2198 | 1,739 | 454.5 | 434.3 | **48.67** |
| | 0.3 | 0.3372 | 1,968 | 452.0 | 438.8 | **50.61** |
| | 0.4 | 0.5510 | 1,759 | 457.5 | 442.4 | **51.21** |
| | 0.5 | 0.08732 | 1,752 | 477.2 | 449.8 | **53.01** |
| Dermatology | 5.0 | 0.5568 | 706.8 | 138.8 | 124.3 | **86.81** |
| | 5.1 | 0.5714 | 654.0 | 142.2 | 127.8 | **64.62** |
| | 5.2 | 0.5733 | 651.8 | 139.2 | 123.2 | **63.74** |
| | 5.3 | 0.5796 | 650.7 | 138.2 | 121.5 | **60.75** |
| | 5.4 | 0.4495 | 638.6 | 138.4 | 121.2 | **58.17** |
| Ecoli | 0.5 | 0.1251 | 506.9 | 116.0 | 104.9 | **7.674** |
| | 0.6 | 0.2820 | 491.9 | 116.0 | 105.2 | **8.105** |
| | 0.7 | 0.3609 | 496.0 | 116.5 | 107.2 | **9.263** |
| | 0.8 | 0.4374 | 500.9 | 116.7 | 105.1 | **10.97** |
| | 0.9 | 0.1563 | 499.8 | 116.3 | 105.0 | **11.39** |
| Phoneme | 8.5 | 0.5142 | 3,497 | 17,290 | 7,685 | **926.9** |
| | 8.6 | 0.5516 | 3,511 | 17,480 | 7,738 | **954.1** |
| | 8.7 | 0.5836 | 3,300 | 17,490 | 7,727 | **937.9** |
| | 8.8 | 0.6028 | 3,257 | 17,600 | 7,768 | **975.2** |
| | 8.9 | 0.5011 | 3,499 | 17,570 | 7,734 | **1,065** |
| Wine | 2.2 | 0.4191 | 73.37 | 64.02 | 56.70 | **5.753** |
| | 2.3 | 0.4764 | 64.65 | 63.84 | 56.29 | **5.703** |
| | 2.4 | 0.5271 | 66.91 | 63.26 | 55.74 | **5.612** |
| | 2.5 | 0.08443 | 67.29 | 63.11 | 56.34 | **6.106** |
| | 2.6 | 0.07886 | 67.12 | 63.45 | 56.25 | **6.094** |

**Note:**
The DBSCAN radius parameter is `eps` and the achieved normalized mutual information is NMI. Best runtimes are highlighted in bold.

`eps` has the same interpretation as SNN's radius parameter $R$. In all cases, we have fixed DBSCAN's second hyperparameter `min_sample` at 5. The normalized mutual information (NMI) (*Cover & Thomas, 2006*) of the obtained clusterings is also listed in Table 7.

The runtimes in Table 7 show that DBSCAN with SNN is a very promising combination, consistently outperforming the other combinations. When compared to using non-batched and non-parallelized DBSCAN with balltree, DBSCAN with SNN performs between 3.5 and 16 times faster while returning precisely the same clustering results.

## CONCLUSIONS

We presented a fixed-radius nearest neighbor (NN) search method called SNN. Compared to other exact NN search methods based on $k$-d tree or balltree data structures, SNN is trivial to implement and exhibits faster index and query time. We also demonstrated that SNN outperforms different implementations of brute force search. Just like brute force search, SNN requires no parameter tuning and is straightforward to use. We believe that SNN could become a valuable tool in applications such as the MultiDark Simulation (*Klypin et al., 2016*) or the Millennium Simulation (*Boylan-Kolchin et al., 2009*). We also demonstrated that SNN can lead to significant performance gains when used for nearest neighbor search within the DBSCAN clustering method.

While we have demonstrated SNN speedups in single-threaded computations on a CPU, we believe that the method's reliance on high-level BLAS operations makes it suitable for parallel GPU computations. A careful CUDA implementation of SNN and extensive testing will be subject of future work.

### Funding
Stefan Güttel's work was supported by a Royal Society Industry Fellowship IF/R1/231032. The funders had no role in study design, data collection and analysis, decision to publish, or preparation of the manuscript.

### Grant Disclosures
The following grant information was disclosed by the authors:
Royal Society Industry Fellowship: IF/R1/231032.

### Competing Interests
The authors declare that they have no competing interests.

### Author Contributions
- Xinye Chen conceived and designed the experiments, performed the experiments, analyzed the data, performed the computation work, prepared figures and/or tables, and approved the final draft.
- Stefan Güttel conceived and designed the experiments, performed the experiments, analyzed the data, performed the computation work, authored or reviewed drafts of the article, and approved the final draft.

## Data Availability

The raw data and code are available at Figshare: Güttel, Stefan; Chen, Xinye (2023). snn_exp. figshare. Dataset. https://doi.org/10.6084/m9.figshare.24781473.v1.

The code is available at GitHub and Zenodo:

- https://github.com/nla-group/snn/.

- the null & Stefan Güttel. (2023). nla-group/snn: v1.0. Zenodo. https://doi.org/10.5281/zenodo.10275014.

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
