# Peer review of "Fast and exact fixed-radius neighbor search based on sorting"

_PeerJ Computer Science, doi:10.7717/peerj-cs.1929_

## Round 0.1 · original submission · Minor Revisions

The two reviewers agree that the manuscript is clear and well-written. The second reviewer poses a number of questions – could you please have a look at their attached PDF, respond to their queries, and update the manuscript as required?

Reviewer 1 ·

Basic reporting

This manuscript was well-written, displaying both clarity and a logical organization of its content. Sufficient background information was provided, making it accessible even to readers who may not have an extensive mathematical background like myself. I found the manuscript to be quite understandable and engaging. However, there are a few typos that need to be corrected.
line 68: "results to do not" shall be "results do not".
line 106: "satisfy ||pi-q||2>R" shall be "satisfy ||pi-q||>R".
line 126: "the smallest such set" shall be "the smallest subset".

Experimental design

The research question was clearly defined. The algorithm was described and explained thoroughly. The theoretical analysis and experimental validations were designed appropriately to investigate the performance of the proposed SNN method.

Validity of the findings

The results were presented with sufficient accuracy, along with high-quality figures and tables, clearly demonstrating the advantages of the proposed SNN method over existing ones. Meanwhile, I'd like to see simulations of datasets with d>>n, i.e., the number of features is larger than the number of samples, which is commonly seen in genomics studies. Comparison of various methods on these datasets will help assess the usefulness of these methods in bioinformatics analysis.

Additional comments

Line 160, the meaning of "overestimation" is unclear. Shall it be "underestimation", or does it refer to |ai-aq|+2sigma2^2?

Cite this review as

Reviewer 2 ·

Basic reporting

I think the authors have carefully crafted the article, no area where they don't meet the standards of the journal.

Experimental design

It does fullfill the standard of a scientific research.

Validity of the findings

No comments

Additional comments

Just need to address a few ambiguities that I mentioned the critical arguments.

Annotated reviews are not available for download in order to protect the identity of reviewers who chose to remain anonymous.
Cite this review as

---

## Round 0.2 · accepted · Accept

I have now received comments on the revised manuscript. The first reviewer highlights that one of their comments has not been addressed in the rebuttal. At first, I couldn't find that comment myself, and that's because it's in the "Validity of the findings" section of the review rather than in the attached PDF document.

I agree with the reviewer that datasets with more features than data points are challenging and that they represent an interesting avenue for future research. On the other hand, I am not entirely convinced that adding an example of such datasets would be within the scope of this particular work.

Therefore, I believe that the manuscript is ready for publication.

Reviewer 1 ·

Basic reporting

The authors have addressed my concerns in Basic Reporting.

Experimental design

Satisfactory.

Validity of the findings

The authors did not address my comment about datasets with d>>n. My original comment is copied here - "Meanwhile, I'd like to see simulations of datasets with d>>n, i.e., the number of features is larger than the number of samples, which is commonly seen in genomics studies. Comparison of various methods on these datasets will help assess the usefulness of these methods in bioinformatics analysis.".

Cite this review as

Reviewer 2 ·

Basic reporting

No comment

Experimental design

No comment

Validity of the findings

No comment

Additional comments

No comment

Cite this review as